# A Mechanistic Model of Perceptual Binding Predicts That Binding Mechanism Is Robust against Noise

**DOI:** 10.3390/e26020133

**Published:** 2024-01-31

**Authors:** Pavel Kraikivski

**Affiliations:** Division of Systems Biology, Academy of Integrated Science, Virginia Polytechnic Institute and State University, Blacksburg, VA 24061, USA; pavelkr@vt.edu

**Keywords:** theory of consciousness, binding problem, consciousness, perceptual binding, perception, neural correlates of consciousness, spectral entropy, power spectrum, stochastic modeling, noise in neuronal networks

## Abstract

The concept of the brain’s own time and space is central to many models and theories that aim to explain how the brain generates consciousness. For example, the temporo-spatial theory of consciousness postulates that the brain implements its own inner time and space for conscious processing of the outside world. Furthermore, our perception and cognition of time and space can be different from actual time and space. This study presents a mechanistic model of mutually connected processes that encode phenomenal representations of space and time. The model is used to elaborate the binding mechanism between two sets of processes representing internal space and time, respectively. Further, a stochastic version of the model is developed to investigate the interplay between binding strength and noise. Spectral entropy is used to characterize noise effects on the systems of interacting processes when the binding strength between them is varied. The stochastic modeling results reveal that the spectral entropy values for strongly bound systems are similar to those for weakly bound or even decoupled systems. Thus, the analysis performed in this study allows us to conclude that the binding mechanism is noise-resilient.

## 1. Introduction

A mechanism that provides a unified conscious representation of a scene that is characterized by different perceptual features is known as perceptual binding [1,2,3]. Thus, the primary function of the binding mechanism is to unify the sensory information processed in different parts of the brain to give us a unitary conscious experience of an object or scene. Several mechanisms have been proposed to solve the binding problem. Temporal neuronal synchrony models propose that different perceptual features are bound together when the firing activities of neurons processing these features are synchronized [2,3,4,5]. Similarly, the temporo-spatial theory of consciousness (TTC) suggests that temporal alignment permits binding between a stimulus and ongoing spontaneous neural activity [6,7]. Operational Architectonics suggests that binding is achieved with operational synchrony among neuronal processes occurring in different brain regions [8,9]. Alternatives to temporal synchrony have also been proposed [10]. In this work, a stochastic mechanistic model of binding is developed and presented based on my previous works [11,12]. The model enables quantification of the interplay between noise and binding.

Noise in neurons may generate significant fluctuations in neuronal responses [13,14], yet sensory features represented by neuronal circuits remain stable [15]. For example, noise affects neuronal signals transmitted by the sensory-motor system [14], operation of voltage-gated channels [16,17], synaptic activity [18,19], potential differences across nerve cell membranes [20], propagation of action potentials [21], and spike train coding [22]. Furthermore, noise can change information processing in sub-threshold periodic signals by helping these signals cross the threshold. Such noise-induced transmission of information has been detected in sensory neurons [23] and mechanoreceptor cells [24,25]. The information capacity of neuronal networks also depends on noise [26]. Additive noise can increase the mutual information of threshold neurons [26,27,28]. Nevertheless, very little is known about how phenomenal states and perceptual binding remain robust against noise despite ubiquitous noise sources in neural circuits. In my previous work, I showed that entropy decreases with an increase in the size of a network that contains negative feedback loops interconnecting processes [12]. In this study, I investigate the interplay between noise and binding strength using the same framework, in which bound phenomenal states are encoded in relationships among processes. A phenomenal state (quale) is postulated to be a dynamic property of running processes, and is isomorphic to the executed relationships among the processes [29,30].

Other theories of consciousness have also postulated that the emergence of a conscious experience is associated with a specific action or execution performed by the brain. The theory of neuronal group selection (TNGS) postulates that qualia are high-dimensional discriminations of specific conscious scenes among a vast repertoire of different possible conscious scenes [31], and differences in qualia are determined by differences in neural structure and dynamics. Similarly, according to the integrated information theory (IIT), qualia arise from the reduction of uncertainty when a particular conscious state occurs out of a repertoire of alternative states [32]. Complex systems with larger numbers of possible states generate more information by reducing uncertainty and, thus, generate complex and vivid conscious experiences. For example, consider the conscious experience of a spatial position of a point-like object (i.e., without shape or any other features except the location) in empty space. According to IIT, the conscious experience of the point location occurs when the brain reduces uncertainty by ruling out all possible different positions of that point in space. However, within this framework, it is not clear how the reduction of uncertainty can reoccur contentiously in time when the phenomenal state is retained in consciousness over time. By contrast, per the dynamical framework presented in my study, the continuous execution of processes is an inherent attribute of the framework. A phenomenal state arises from the execution of relationships among processes and is then isomorphic to the executed relationships. Therefore, the phenomenal state is a dynamic property that exists as long as the execution of this property by the system continues in time. In the above example with the spatial position, the phenomenal state isomorphic to a specific position in space would arise when the relationships among a process assigned to the specific position and other processes assigned to all other possible locations in space are executed. Furthermore, the phenomenal state, in this case, would represent not a single point by itself, but the point within the internal phenomenal space.

In this work, I present a mechanistic model that describes perceptual binding between a system’s encoded space and time, which are isomorphic to Euclidean space and time, respectively. The same framework can also be applied to other examples of perceptual binding [11]. The main goal of this work is to investigate how binding is affected by noise. The implications of noise for the system are quantified using spectral entropy, and the results indicate that the binding mechanism is robust against noise.

## 2. Materials and Methods

The system of oscillating processes, along with the relationships among processes, are used to represent the physical carrier of phenomenal states. The system’s internal representations of space and time are assumed to be encoded in relationships among the processes that are described by the following variables: $\vec{P(t)}=(p_{1}(t),\;p_{2}(t){,\;\ldots ,\;p}_{n}(t))$ and $\vec{Q(t)}=(q_{1}(t),\;q_{2}(t){,\;\ldots ,\;q}_{m}(t))$, where t is regular external time. There are *n* number of oscillating processes to encode space and *m* number of processes to encode time. The internal space and time are encoded in the relationships among processes, which describe how the processes influence each other or interact. For the brain or neural networks, such relationships would be set by entrainment with external stimuli that are placed at different spatial positions and act with varying time intervals. I assume that the system of processes is already entrained and, thus, each process has specific relationships to all other processes as $\vec{P}=A\vec{P}$ and $\vec{Q}=B\vec{Q}$, where the structure of ***A*** and ***B*** matrices represents memory, which should be isomorphic to real space and time. The elements of ***A*** and ***B*** matrices are independent of time. However, the relationships among processes encoded in ***A*** and ***B*** are continuously executed as all processes continuously oscillate in time. This is an important concept of this framework, where a phenomenal state (quale) is assumed to be a property of a dynamical system, which emerges and exists when that property is realized or “happens”. Therefore, the relationships among processes must be continuously executed, yet the specific relationships among processes must be maintained over time, as long as the experience of the corresponding phenomenal state is unchanged. Although, in general, the relationships among processes can be nonlinear (e.g., the relationships between two processes which are represented by a limit cycle in the phase plane), here, I assume that phenomenal space and time are linear and isomorphic to Euclidean space. Therefore, the Euclidean distance hollow matrices below:(1)$$A=\left(\begin{array}{cccc} 0 & \varepsilon & \cdots & ({n-1)}^{2}\varepsilon \\ \varepsilon & 0 & \cdots & ({n-2)}^{2}\varepsilon \\ \vdots & \vdots & \ddots & \vdots \\ ({n-1)}^{2}\varepsilon & ({n-2)}^{2}\varepsilon & \cdots & 0 \end{array}\right)$$
(2)$$\mathrm{and}\;B=\left(\begin{array}{cccc} 0 & \alpha & \cdots & ({m-1)}^{2}\alpha \\ \alpha & 0 & \cdots & ({m-2)}^{2}\alpha \\ \vdots & \vdots & \ddots & \vdots \\ ({m-1)}^{2}\alpha & ({m-2)}^{2}\alpha & \cdots & 0 \end{array}\right)$$are used to represent the following relationships among processes: $\vec{P}=A\vec{P}$ and $\vec{Q}=B\vec{Q}$. Thus, for the $p_{i}$ and $q_{i}$ components of $\vec{P}$ and $\vec{Q}$, we can also write:(3)$$p_{i}=\sum_{j=1}^{n}{\left(i-j\right)}^{2}\varepsilon p_{j}$$
(4)$$\mathrm{and}\;q_{i}=\sum_{j=1}^{m}{\left(i-j\right)}^{2}\alpha q_{j},$$
where ε and α are scaling parameters for the “distance and interval” measures between processes. If the sets of processes $\vec{P}$ and $\vec{Q}$ that describe the internal representations of space and time are not coupled, then their dynamics are described by the following systems of ordinary differential equations:(5)$$\;\frac{d\vec{P}}{dt}=A\vec{P}-\left(\vec{X}+\vec{P}\right)$$
(6)$$\frac{d\vec{X}}{dt}=\vec{P}\;\;\;\;\;\;\;\;\;\;\;\;\;\;\;\;\;\;\;\;\;\;\;\;\;\;\;\;\;\;\;\;\;$$
(7)$$\;\frac{d\vec{Q}}{dt}=B\vec{Q}-\left(\vec{Z}+\vec{Q}\right)$$
(8)$$\frac{d\vec{Z}}{dt}=\vec{Q},\;\;\;\;\;\;\;\;\;\;\;\;\;\;\;\;\;\;\;\;\;\;\;\;\;\;\;\;\;\;$$

The systems of Equations (5)–(8) have the following analytical solutions: $\vec{P}=A\vec{P}$ and $\vec{Q}=B\vec{Q}$ with oscillating $\vec{P}=\vec{K}\cos\left(\lambda t\right)+\vec{L}\sin\left(\lambda t\right)$ and $\vec{Q}=\vec{H}\cos\left(\eta t\right)+\vec{R}\sin\left(\eta t\right)$, where $\vec{K}$, $\vec{L},\;\vec{H},\;$$\vec{R}$ are sets of amplitude values and λ and η are frequencies. Additionally, $\vec{X(t)}=(x_{1}(t),\;x_{2}(t){,\;\ldots ,\;x}_{n}(t))$ and $\vec{Z(t)}=(z_{1}(t),\;z_{2}(t){,\;\ldots ,\;z}_{m}(t))$ are sets of auxiliary processes. If the processes representing internal space and time are bound, then Equations (5) and (7) must be coupled by including the terms that describe an interaction between $\vec{P}$ and $\vec{Q}$ processes. Different coupling schemes were investigated in my previous study [11]. Here, the number of modeled processes is reduced to simplify the stochastic modeling that is used to investigate noise effects on the coupled systems. A dynamical system that contains a set of two processes, $\vec{P(t)}=(p_{1}(t),\;p_{2}(t))$ representing the internal space bound to two processes, and $\vec{Q(t)}=(q_{1}\left(t\right),\;q_{2}\left(t\right))$ representing the internal time, can be described by the following system of coupled equations:(9)$$\frac{dp_{1}}{dt}=\varepsilon p_{2}-p_{1}-x_{1}+\omega f_{1}(q_{1},\;q_{2})$$
(10)$$\frac{dp_{2}}{dt}=\varepsilon p_{1}-p_{2}-x_{2}+\omega f_{2}(q_{1},\;q_{2})$$
(11)$$\;\frac{dx_{1}}{dt}=p_{1}$$
(12)$$\;\frac{dx_{2}}{dt}=p_{2}$$
(13)$$\frac{dq_{1}}{dt}={\alpha q}_{2}-q_{1}-z_{1}+\omega g_{1}(p_{1},\;p_{2})$$
(14)$$\frac{dq_{2}}{dt}=\alpha q_{1}-q_{2}-z_{2}+\omega g_{2}(p_{1},\;p_{2})$$
(15)$$\;\frac{dz_{1}}{dt}=q_{1}$$
(16)$$\;\frac{dz_{2}}{dt}=q_{2}$$

The binding interaction between the $\left(p_{1},\;p_{2},\;x_{1},\;x_{2}\right)$ and $\left(q_{1},\;q_{2},\;z_{1},\;z_{2}\right)$ sets of processes is described by $f_{1}\left(q_{1},\;q_{2}\right)$, $f_{1}\left(q_{1},\;q_{2}\right)$ and $g_{1}\left(p_{1},\;p_{2}\right)$, $g_{1}\left(p_{1},\;p_{2}\right)$ functions. Here, the functions are set as: $f_{1}\left(q_{1},\;q_{2}\right)=q_{1}$, $f_{2}\left(q_{1},\;q_{2}\right)=q_{2}$, $g_{1}\left(p_{1},\;p_{2}\right)=-p_{1}$,and $g_{2}\left(p_{1},\;p_{2}\right)=-p_{2}$. This interaction scheme is shown in Figure 1a. The binding strength between the $p_{i}$ and $q_{i}$ processes depends on the parameter ω. The sign of parameter ε determines whether the $p_{1}$ and $p_{2}$ processes are mutually activating (ε>0) or inhibiting (ε<0) each other. Similarly, the sign of parameter α determines whether the $q_{1}$ and $q_{2}$ processes are mutually activating (α>0) or inhibiting (α<0) each other. This interaction scheme with a fixed coupling constant ω=1 and an alternative wiring—$f_{1}\left(q_{1},\;q_{2}\right)=q_{1}-q_{2}$, $f_{2}\left(q_{1},\;q_{2}\right)=q_{2}-q_{1}$, $g_{1}\left(p_{1},\;p_{2}\right)=p_{2}-p_{1}$, and $g_{2}\left(p_{1},\;p_{2}\right)=p_{1}-p_{2}$—were investigated in my previous study [11]. In this work, the dynamic behavior of the system is analyzed as a function of coupling strength parameter ω. As an example, numerical solutions of Equations (9)–(16) for the $\vec{P(t)}=(p_{1}(t),\;p_{2}(t))$ and $\vec{Q(t)}=(q_{1}\left(t\right),\;q_{2}\left(t\right))$ processes obtained for three different binding strength parameter values (ω= 0.1, 0.5, and 1) are shown in Figure 1b–d.

Next, the system of Equations (9)–(16) is converted into a stochastic model using Gillespie’s method. For the system $\vec{S}=(p_{1},\;p_{2},\;x_{1},\;x_{2},\;q_{1},\;q_{2},\;z_{1},\;z_{2})$, the states are updated using the following general Gillespie’s scheme [33]: Initialize the process state vector, $\vec{S}$, and set the initial time at 0.Calculate the propensities, $a_{k}(\vec{S})$.Generate a uniform random number, $r_{1}$.Compute the time for the next event, $\tau =-\frac{1}{\sum_{k}a_{k}\left(\vec{S}\right)}\ln r_{1}$.Generate a uniform random number, $r_{2}$.Find which event is next, I=i, if $\frac{\sum_{k=1}^{i-1}a_{k}\left(\vec{S}\right)}{\sum_{k}a_{k}\left(\vec{S}\right)}\leq r_{2}<\frac{\sum_{k=1}^{i}a_{k}\left(\vec{S}\right)}{\sum_{k}a_{k}\left(\vec{S}\right)}$Update the state vector, $\vec{S}\to \vec{S}+y_{i}$.Update time, t→t+τ.Repeat steps (2)–(8).


The stochastic model is used to characterize the interplay between the binding strength, ω, and noise. All numerical solutions of Equations (9)–(16) are obtained using XPP/XPPAUT software (http://www.math.pitt.edu/~bard/xpp/xpp.html, accessed on 4 November 2023). The XPP/XPPAUT codes that are sufficient to reproduce all results presented in this work are provided in Appendix A. Code A is used to generate results for the deterministic model described by Equations (9)–(16), and Code B is used to perform stochastic simulations of the model and produce the corresponding model results.

Spectrum analysis and spectral entropy are used to quantify noise effects on the system. Spectral analysis is a common tool in signal processing and in neurophysiological studies [34,35,36,37]. Spectral entropy is based on Shannon’s entropy formalism, which is a foundational concept of information theory [38]. The entropy metric is an important component of the information integration theory of consciousness [39,40]. I have used spectral analysis tools to study noise effects on systems of different sizes, which are described by Equations (5) and (6) [12]. Here, I use the same method to compute spectral entropy for two systems of bound processes (Equations (9)–(16)), in order to characterize the interplay between binding strength and noise.

The spectral entropy value *H* is computed using the following equation:(17)$$H=-k\sum_{j=1}^{2048}\hat{PSD_{j}}Log_{2}({\hat{PSD}}_{j}),\;$$
where $k=\frac{1}{Log_{2}\left(2048\right)}\approx 0.1$ and $\hat{PSD}$ is the normalized power spectral density that is computed by dividing the power spectral density by the total power [41]. The power spectral density is computed from the fast Fourier transform (FFT) obtained for each process trajectory $p_{i}\left(t\right)$ that is simulated using Code B in Appendix A. The Fourier Analysis function in Excel’s Analysis ToolPak is used to obtain the corresponding signal $p_{i}\left(f\right)$ in the frequency domain. 4096 points are used to compute $p_{i}\left(f\right)$, which corresponds to a total average simulation time of ~930 arb. u. where the period of oscillations ranges between ~4–7 arb. u. The sampling frequency, f, is obtained by dividing the number of points by the time interval, ∆t. The frequency magnitudes are computed using Excel’s IMABS function. The power spectral density is calculated using the following formula: $PSD_{j}={\left|p\left(f_{j}\right)\right|}^{2}/2∆f$. 2048 data points are used to compute spectral densities and the corresponding spectral entropy value from Equation (17). Finally, the coupling parameter ω is varied to characterize the effect of binding strength on the system’s spectral entropy values. For each fixed value of coupling parameter ω, simulations are repeated ten times. Then, those ten spectral entropy values are used to compute the average spectral entropy value and the corresponding standard deviation from the mean.

## 3. Results

In my previous work [11], the deterministic mathematical model described by the system of Equations (9)–(16) was successfully applied to study the perceptual binding between the location of a stimulus at two possible positions and the presence or absence of a light stimulus at these positions. However, the binding strength ω was assumed to be at its largest value, ω=1. It was shown that the system of Equations (9)–(16) exhibits different regimes of modulated oscillations depending on the ε and α parameter values [11]. By contrast, in this study, ε and α are fixed and the binding strength ω is varied. Furthermore, the model is used to describe the possible binding mechanism between encoded space and time. Thus, the model allows one to investigate how the encoding signals may change if the binding strength between the encoded space and time is varied. It is assumed that space and time could be perceived independently as well as together. This assumption follows from the assumption that entrainment of the subsystem that encodes the internal representation of space can be performed either simultaneously with, or independently from, entrainment of the subsystem that encodes the internal representation of time. In addition, stochastic simulations of the model are performed to characterize the robustness of the binding mechanism against noise.

When two oscillatory systems are coupled, they modulate each other and, thus, the oscillatory dynamics of the coupled system are altered. The modulation depends on the parameter ω, that describes the binding strength between two oscillatory systems. This is seen in Figure 1a. Figure 1b–d demonstrate how the binding strength parameter influences the modulation of two coupled oscillatory systems. To characterize the robustness of the coupled system against noise, stochastic simulations are performed. The results of the stochastic model simulations are shown in Figure 2. The stochastic model is simulated using different values of the binding strength parameter ω. Figure 2a,d,g present the stochastic trajectories for the $\vec{P(t)}=(p_{1}(t),\;p_{2}(t))$ and $\vec{Q(t)}=(q_{1}\left(t\right),\;q_{2}\left(t\right))$ processes obtained for the binding strength parameter values ω = 0.1, 0.5, and 1. These results can be compared to the numerical results of the deterministic model shown in Figure 1b–d, which are obtained for the same ω parameter values; however, the initial conditions used for the simulation results in Figure 1 and Figure 2 are different. Figure 2b,e,h show distribution histograms for the process $p_{1}$, obtained from trajectories recorded over much larger time frames (>1000 arb. u.) than shown in Figure 2a,d,g. Also, for the $p_{2}$ and $q_{1,2}$ processes, the corresponding histograms (not shown) appear similar to those shown in Figure 2a,d,g. Normalized power spectral densities are also computed from trajectories, as described in the Methods section. Alteration of the normalized power spectral densities for the process $p_{1}$ as a function of varied binding strength parameter values is demonstrated in Figure 2c,f,i. There is a shift of normalized power spectrum peaks to higher frequencies as the binding strength ω increases.

Next, spectral entropy is calculated using the power spectra to quantify sensitivity of the coupled systems of processes to noise. For cases shown in Figure 2, spectral entropy values of ≈0.55, 0.57, and 0.55 are obtained using the power spectra shown in Figure 2c,f,i, with binding strength parameter values ω= 0.1, 0.5, and 1, respectively. Because the spectral entropy values do not change significantly with alteration of the binding strength parameters, it can be concluded that strongly and weakly bound oscillatory systems are equally robust against noise. To test this hypothesis in a more systematic way, fifty independent simulation experiments for the coupled systems of processes are performed, with ten simulation experiments for each of five different values of the binding strength parameter: ω = 0, 0.25, 0.5, 0.75, and 1. In each independent simulation, the spectral entropy value is computed. Then, the average over ten spectral entropy values is calculated for each specific ω parameter value. Figure 3 shows the average spectral entropy values plotted versus the binding strength parameter. The error bars represent standard deviation values. The results indicate that the robustness of the coupled systems against noise does not vary significantly when the binding strength changes. Therefore, the binding mechanism used to couple two oscillatory systems is resilient to noise.

## 4. Discussion

The brain’s ability to construct its own space and time provides an essential foundation for conscious processing of the outside world. All other phenomenal aspects are built upon this foundation. When we have perceptual experiences of different phenomenal aspects, such as those related to our perceptions of colors, odors, tactile features, etc., they are always inseparably unified with our internal representations of space and time. Thus, a solution to the perceptual binding problem must include phenomenal space and time as the common foundation that unifies other phenomenal aspects into a single experience.

In this work, I present a mechanistic stochastic model of perceptual binding between encoded space and time. Because variations in spatial patterns and temporal changes could, in principle, be perceived as separable events, I assume that phenomenal representations of space and time can be unbound, or weakly or strongly bound. Thus, the mechanistic model of binding is used to investigate how the oscillating processes that encode the internal (phenomenal) representations of space and time are modulated when the binding strength between them is varied. Furthermore, stochastic simulations of the model are used to analyze the interplay between the binding strength and noise. The model results suggest that the binding mechanism is robust against inherent noise. Therefore, the model provides some explanation as to why perceptual experiences and perceptual binding can be robustly retained and unchanged despite ubiquitous noise sources in neuronal circuits. In my previous work, it was shown that large systems involving more interconnected oscillating processes are less noise sensitive than small systems with fewer processes [12]. Noise is suppressed in large systems by negative feedback loops that are involved in a network of interconnected processes. The peaks of power spectral densities were shifted from low to high frequency values with an increase in the number of processes [12]. Figure 2c,f,i also show a shift in normalized power spectrum peaks to higher frequencies as the binding strength increases. This shift to higher frequencies occurs because the binding mechanism involves negative feedback loops, as seen in Figure 1a. Similar observations have been reported for gene regulatory networks with negative feedback loops [42,43]. Moreover, it has also been observed that noise-like signals associated with scale-free brain activity may play an important role in determining the state of consciousness [44]. Therefore, an important future direction would be to apply the mechanistic model to investigate the implications of scale-free dynamics on perceptual binding.

In this study, the interplay between binding strength and noise is characterized using power spectral density and spectral entropy values. Spectral analysis has often been used to analyze electroencephalograms to study the neurophysiology of sleep [34], predict changes in memory performance [41], and detect differences in brain activities of subjects under different conditions [35,36,37]. Because the same spectral analysis tools are used in this work, it should be relatively easy to compare results and validate conclusions derived from the model simulations with results obtained in neurophysiological experiments.

However, it should be noted that the oscillating processes described by the model cannot be explicitly related to membrane potentials and the spiky oscillations exhibited by individual neurons. The dynamics of a spiking neuron can be better represented by the Hodgkin-Huxley and FitzHugh–Nagumo models, which employ nonlinear differential equations [45,46,47,48]. However, the nonlinear neuronal impulses have complex relationships that are not isomorphic with Euclidian space and time, which are the subjects of this study. The processes described by my model can be attributed to the dynamics of neural populations [49]. For example, averaged evoked potentials (AEP) recorded from different parts of the brain using electroencephalography (EEG) are well-fitted using sinusoidal functions [50,51]. Linearized approximation has been successfully applied to describe cortical evoke potentials [50,51,52,53]. Thus, the model results can be compared with neural population responses recorded by EEG techniques. Some oscillating patterns of the EEG with varying amplitudes are similar to the oscillation patterns obtained in this work (compare Figure 2a in this work with Figure 3 on page 33 in Ref. [50]). Although many EEG signals appear complex and noisy, the principle of superposition can be applied to separate the complex composite signals into components [52]. Then, in line with the hypothesis employed in my model, the oscillating electric field components that form relationships isomorphic with a conscious percept can contribute to the conscious state. It is possible that no “meaningful” contribution can emerge from electric field components that do not retain the relationships isomorphic with the percept.

It should also be noted that the internal representations of space and time modeled in this work are different from the inner space and time postulated in the temporo-spatial theory of consciousness (TTC) [6,7]. The inner time in TTC is related to the temporal ranges of neural oscillations that arise in different forms of neural activity. The inner space is related to spatial ranges of neural activity across different regions in the brain. Therefore, inner space and time in TTC are constructed using characteristic spatial ranges and timescales in different forms of neural activity. The inner space and time in TTC are, thus, different from the internal phenomenal representations of space and time. Furthermore, TTC suggests that binding between different forms of neural activity is determined by “temporo-spatial alignment”. Temporo-spatial alignment dictates whether different forms of neural activity and their respective contents can be merged and associated with consciousness. Importantly, the temporal integration of different forms of neural activity is based on their temporal properties and is independent of their specific contents. Thus, TTC highlights the difference between binding based on the temporal alignment and a content-based integration. Similarly, the binding mechanism in my model concerns interactions among oscillating processes. The mechanism permits the mutual modulation of processes regardless of specific phenomenal content carried by the processes. However, content integration is governed by relationships among processes and their changes occurring due to interactions.

Overall, the mechanistic model allows us to better understand how binding can alter dynamics of neural-like oscillatory systems. The model results can help to interpret some neural population activity patterns as recorded by EEG techniques. Furthermore, the model can be applied to describe the binding mechanism between any two percepts that can be represented by two systems of oscillating processes, as has been demonstrated in my previous work [11]. The stochastic version of the model gives us a useful tool to study noise effects on systems that involve binding. It can also be used to investigate mechanisms with which the brain suppresses or employs inherent noise to make our perceptual binding and experiences sturdy.

## Figures and Tables

**Figure 1 entropy-26-00133-f001:**
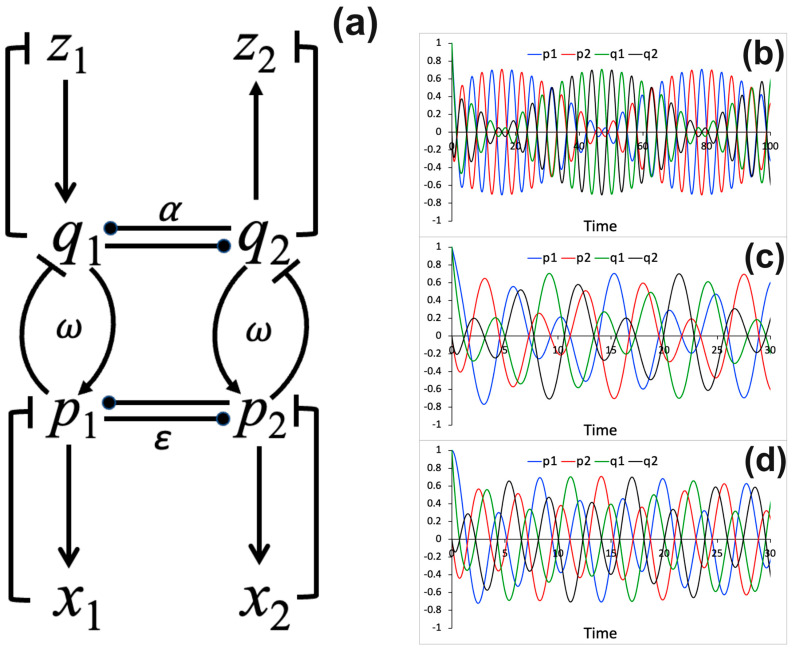
The interaction diagram and dynamical relationships among processes. (**a**) A diagram that shows interactions among processes. (**b**–**d**) Numerical solutions for the time evolution of $\vec{P(t)}=(p_{1}(t),\;p_{2}(t))$ and $\vec{Q(t)}=(q_{1}\left(t\right),\;q_{2}\left(t\right))$ processes computed using different values of the coupling constant: (**b**) ω=0.1, (**c**) ω=0.5, (**d**) ω=1. In all simulations, ε=−1 and α=−1, thus, representing mutual inhibition between the $p_{1}$ and $p_{2}$ processes and between the $q_{1}$ and $q_{2}$ processes. The following initial conditions are used: $\vec{P(0)}=(1,\;0)$, $\vec{X(0)}=(0,\;0)$, and $\vec{Q(0)}=\left(1,\;0\right)$, $\vec{Z(0)}=(0,\;0)$.

**Figure 2 entropy-26-00133-f002:**
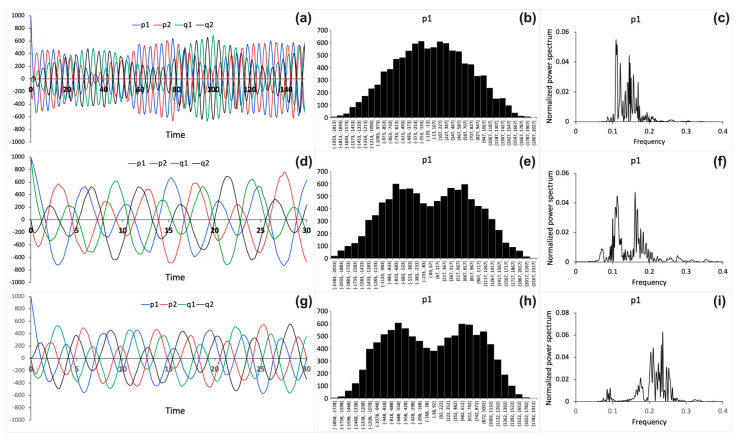
Numerical stochastic simulation results obtained using the following binding strength parameter values: (**a**–**c**) ω= 0.1, (**d**–**f**) ω= 0.5, and (**g**–**i**) ω= 1. (**a**,**d**,**g**) Stochastic trajectories for the $\vec{P(t)}=(p_{1}(t),\;p_{2}(t))$ and $\vec{Q(t)}=(q_{1}\left(t\right),\;q_{2}\left(t\right))$ processes. Time is shown in arbitrary units. The following initial conditions are used: $\vec{P(0)}=(1000,\;0)$, $\vec{X(0)}=(0,\;0)$, $\vec{Q(0)}=\left(1,\;0\right)$, $\vec{Z(0)}=(0,\;0)$ in (**a**,**g**) and $\vec{P(0)}=(1000,\;0)$, $\vec{X(0)}=(0,\;0)$, $\vec{Q(0)}=\left(1000,\;0\right)$, $\vec{Z(0)}=(0,\;0)$ in (**d**). (**b**,**e**,**h**) Distribution histograms for process $p_{1},$ computed using trajectories recorded over (**b**) 1580 arb. u., (**e**) 1120 arb. u., and (**h**) 1071 arb. u time frames. (**c**,**f**,**i**) Normalized power spectral densities for process $p_{1}$.

**Figure 3 entropy-26-00133-f003:**
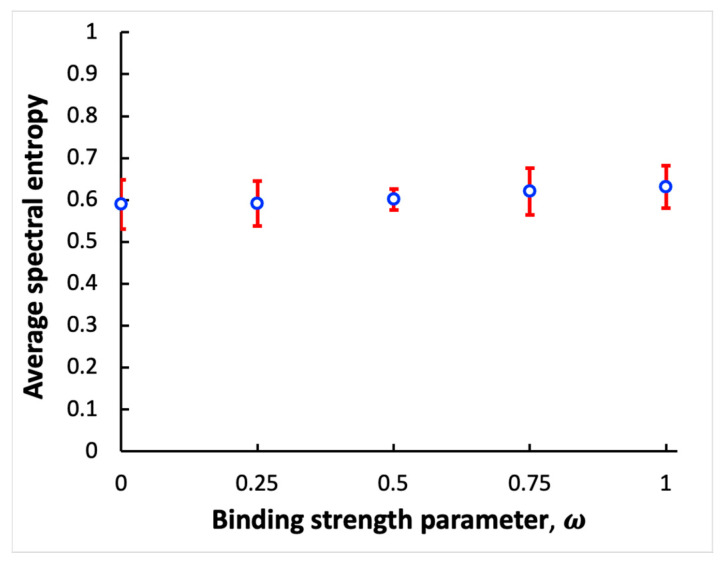
Dependence of spectral entropy on the coupling strength between two bound oscillatory systems. Open circles represent the average spectral entropy values obtained using different values of the binding strength parameter ω. Error bars provide standard deviation values.

## Data Availability

The data presented in this study are available on request from the corresponding author.

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
