# Peer review of "A Mechanistic Model of Perceptual Binding Predicts That Binding Mechanism Is Robust against Noise"

_entropy, 2024, doi:10.3390/e26020133_

Round 1

Reviewer 1 Report

Comments and Suggestions for Authors

Based on modeling two systems made of oscillating processes, the author argues that the strength of the coupling of inner space and time is resilient to noise.

Comments: It is difficult to refer to equations as they are not individually labeled as equations # 1, 2, 3, etc.

It is unclear how space and time differ based on equations and mathematical descriptions (lines 109-123).

It appears that oscillating processes are simple sinusoidal functions that can fail to capture the complexity of the processing of space and time in the brain.

I also find the use of ‘inner space’ ambiguous since the authors are referring to an ‘internal representation of space.’

In lines 115 and 116, the authors argue, “If the processes representing inner space and time are independent, then  their dynamics are described by the following systems of ordinary differential equations:” It is unclear how simply having separate equations can ‘prove’ that the respective equations independent. Nonetheless, if there are other arguments, the authors should clarify them in this paper.

Authors could also add systems of equations on both sides of equality, representing ‘inner’ space and time, to make sense of the effect of noise or spectral entropy on binding strength. If authors had a reason not to combine equations, they must clarify.

Reviewer 2 Report

Comments and Suggestions for Authors

The author presents a mathematical model to study the binding mechanism that unifies different perceptual features into a coherent conscious experience. The model consists of interconnected processes encoding an "inner space" and "inner time", inspired by the temporo-spatial theory of consciousness. The simulations demonstrate the modulation of oscillations in bound systems depends on coupling strength. Spectral analysis quantifies the effects of noise. As binding strength changes, spectral entropy remains similar, indicating the binding mechanism provides potential noise resilience.

Major Concerns:

1. The model makes substantive assumptions about neural encoding of space and time by abstract processes and their relationships. Further justification grounded in neuroscience literature is needed regarding how the model processes relate to actual neural processes and connectivity patterns.

2. Additional interpretation would strengthen connections between key model features (e.g. coupling strength and noise resilience) and the overarching theory of consciousness motivating the work.

Minor Issue:

- The information integration theory should be abbreviated as IIT rather than ITT.
